# PA6 and Halloysite Nanotubes Composites with Improved Hydrothermal Ageing Resistance: Role of Filler Physicochemical Properties, Functionalization and Dispersion Technique

**DOI:** 10.3390/polym12010211

**Published:** 2020-01-15

**Authors:** Valentina Sabatini, Tommaso Taroni, Riccardo Rampazzo, Marco Bompieri, Daniela Maggioni, Daniela Meroni, Marco Aldo Ortenzi, Silvia Ardizzone

**Affiliations:** 1Dipartimento di Chimica, Università degli Studi di Milano, Via Golgi 19, 20133 Milano, Italy; tommaso.taroni@unimi.it (T.T.); riccardo.rampa@gmail.com (R.R.); marco.bompieri@studenti.unimi.it (M.B.); daniela.maggioni@unimi.it (D.M.); daniela.meroni@unimi.it (D.M.); marco.ortenzi@unimi.it (M.A.O.);; 2Consorzio Interuniversitario per la Scienza e Tecnologia dei Materiali (INSTM), Via Giusti 9, 50121 Firenze, Italy; 3CRC Materiali Polimerici “LaMPo”, Dipartimento di Chimica, Università degli Studi di Milano, Via Golgi 19, 20133 Milano, Italy

**Keywords:** polyamide 6, halloysite nanotube, functionalizing agent, nanocomposite, *in situ* polymerization, melt blending, polymorphism, hydrothermal ageing

## Abstract

Polyamide 6 (PA6) suffers from fast degradation in humid conditions due to hydrolysis of amide bonds, which limits its durability. The addition of nanotubular fillers represents a viable strategy for overcoming this issue, although the additive/polymer interface at high filler content can become privileged site for moisture accumulation. As a cost-effective and versatile material, halloysite nanotubes (HNT) were investigated to prepare PA6 nanocomposites with very low loadings (1–45% w/w). The roles of the physicochemical properties of two differently sourced HNT, of filler functionalization with (3-aminopropyl)triethoxysilane and of dispersion techniques (*in situ* polymerization vs. melt blending) were investigated. The aspect ratio (5 vs. 15) and surface charge (−31 vs. −59 mV) of the two HNT proved crucial in determining their distribution within the polymer matrix. *In situ* polymerization of functionalized HNT leads to enclosed and well-penetrated filler within the polymer matrix. PA6 nanocomposites crystal growth and nucleation type were studied according to Avrami theory, as well as the formation of different crystalline structures (α and γ forms). After 1680 h of ageing, functionalized HNT reduced the diffusion of water into polymer, lowering water uptake after 600 h up to 90%, increasing the materials durability also regarding molecular weights and rheological behavior.

## 1. Introduction

In recent decades, polyamide 6 (PA6) has received considerable attention [1] for a broad range of applications including in the automotive sector [2] and the textile industry [3], owing to its good mechanical performances and high thermal resistance, which are the result of the strong hydrogen bonds in its chemical structure. 

However, in demanding environmental conditions (i.e., extremely high/low temperatures, high humidity, solar exposure and salts absorption), different degradation mechanisms occur and decrease PA6 material performance and durability [4,5,6]. In particular, in humid conditions, water can interpenetrate between –NH and C=O groups of PA6 and break the pre-existent bonds, leading to a loss in mechanical cohesion [7,8]. Because the prediction and extension of PA6 durability is a very important issue for chemists, engineers and final users, numerous studies have been devoted to investigate its hydrothermal ageing. 

In this context, PA6 properties can be preserved or modified to suit engineering requirements for several applications, using fillers and fiber additives such as silicate clays [9], glass fiber [10], black carbon [11], graphite [12], silica [13] and graphene [14]. However, it should be underlined that in some composites, such as PA6 and glass fibers-based materials, the additive/matrix interface is considered a privileged site for water accumulation during diffusion processes. Since this behavior can become catastrophic as the additive content increases [15,16], the use of low filler amounts represents a preferable strategy. 

By using fillers with nanotubular shapes, significant improvements in the composite properties can be obtained using very low loadings, even down to a range of 1–5% w/w, maintaining in this way processing conditions similar to the ones of the neat material [9,17,18]. This is because their high surface/volume ratio favors their dispersion into the polymer matrix. Thus far, numerous investigations have focused on PA6 functionalization with carbon nanotubes (CNTs) [19,20,21] and boron nitride nanotubes (BNNTs) [22,23], often with very high loadings (up to 40% w/w [24,25]), due to the interesting mechanical properties of the resulting nanocomposites. However, both of these additives are technologically demanding to produce in bulk and functionalize, making them quite expensive and suitable only for limited high added value applications, e.g., robotics and organic electronics [26,27]. Thus, there is still a need to develop new types of high-performance PA6-nanocomposites that can be prepared via large scale, low cost fabrication routes.

In this framework, a phyllosilicate clay, halloysite nanotubes (HNT), represents a novel 1D natural nanomaterial morphologically similar to CNTs and BNNTs, with a unique combination of tubular nanostructure, natural availability, low cost, ease of functionalization, good biocompatibility, light color and high mechanical strength [28,29,30]. These features can impart useful mechanical, thermal, and biological properties to HNT-polymer nanocomposites, while keeping costs low [31,32]. Nanocomposites of polyamides and HNT have been reported in the literature to exhibit higher thermal stability [33] and flame inhibition properties [34], as well as enhanced strength without loss of ductility [35], even at very low filler content (<5%) [36,37,38]. However, to the authors’ best knowledge, no previous reports about the effect of HNT filler on the durability of polyamide materials in hydrothermal conditions can be found in the literature. The present study aims at shedding light on this aspect by exploiting the effects of low content (1% and 4% w/w) of HNT fillers on the hydrothermal ageing of PA6 nanocomposites.

As halloysite is an extractive material, it shows a large morphological variance (from dimension to specific surface area and aspect ratio) depending on the extraction site. This aspect can have a great impact on the final properties of nanocomposites, but it has often been overlooked in previous literature reports. In this regard, here we thoroughly compare two different sources of HNT with several structural, morphological, optical and surface differences. 

Moreover, we here compare the two different incorporation strategies more often used for fillers incorporation in PA6: in situ polymerization and melt blending processes. In situ polymerization involves modifiers dispersion into monomer followed by polymerization, while melt extrusion proceeds via a mechanical blending of filler in the molten polymer. The melt blending approach is more cost effective and straightforward than the in situ polymerization technique, but full filler intercalation is not always achievable [39]. On the other hand, relatively little work has been reported on the in situ polymerization technique, particularly concerning the relationship between the filler dispersion efficiency and PA6-based composites resultant macromolecular properties [39], and no previous work can be found in relation to HNT-polyamide composites using this incorporation strategy. 

One of the main advantages of HNT is their ease of functionalization, which enables the tailoring of their surface properties. Guo et al. [33] reported the covalent incorporation of HNT functionalized with 3-(trimethoxysilyl)propyl methacrylate within a PA6 matrix, showing good incorporation and enhanced thermal and mechanical properties of the resulting composite. Here we compare HNT fillers both in their pristine state and after functionalization with (3-aminopropyl)triethoxysilane (APTES); this surface modification was adopted to increase compatibility between the filler surface and the organic matrix, as well as to reduce the surface hydrophilicity. 

For all of the prepared composites, the effect of HNT incorporation on the composite molecular weights, molecular weight distribution, thermal and rheological properties, together with the filler dispersion state inside the polymer matrix, was investigated with respect to standard PA6. In particular, the comparison between in situ and melt blending methods highlighted the role of HNT surface charge with respect to the filler dispersion efficiency. Finally, the composite resistance to humidity was determined via accelerated ageing tests. It is well known that the macroscopic properties of most polyamides are affected by hydrothermal ageing after few months, or even weeks [40,41,42]. In the present case, notable differences between the neat PA6 and the composite materials are observed even after just 70 days.

## 2. Materials and Methods

### 2.1. Materials

ε-caprolactam (99%), 6-aminocaproic acid (≥99%), distilled water Chromasolv^®^ (≥99.9%), toluene (anhydrous, ≥99.8%), potassium nitrate (KNO_3_), 2-propanol, (3-aminopropyl)triethoxysilane (APTES, ≥98%) and dichloromethane (DCM anhydrous, ≥99.8%) were supplied by Sigma Aldrich (Milan, Italy and used without further purification. The commercial polyamide 6 “Technyl^®^ C 246 natural” (PA6TECH) was purchased from Solvay Chimica Italia S.p.A (Livorno, Italy). Two different kinds of HNT nano-clays were used to prepare nanocomposites: HNT_H_ supplied by iMineral Inc. (commercially known as Ultra HalloPure^®^, Vancouver, Canada) and HNT_S_ received from Sigma Aldrich (Milan, Italy). 

### 2.2. HNT Functionalization

2 g of HNT was suspended in 100 mL of toluene. Then, 1.4 g (6.3 mmol) of APTES was added and the suspension was refluxed under stirring for 7 h. Heating was stopped during the night (15 h) and the retrieved powder was washed with toluene by centrifugation/resuspension cycles (5 × 40 mL). The solid was then dried in the oven at 90 °C for 24 h.

### 2.3. HNT Characterization

The specific surface area (SSA) of the samples was measured through N_2_ adsorption-desorption isotherms in subcritical conditions, utilizing a Coulter SA 3100 apparatus (Beckman Coulter Life Sciences, Indianapolis, Indiana), analyzed according to the Brunauer Emmett Teller (BET) method.

X-ray diffraction (XRD) (Shimadzu, Milan, Italy) patterns of the samples were acquired using a Siemens D5000 diffractometer equipped with a Cu Kα source (λ = 0.15406 nm), working at 40 kV × 40 mA nominal X-ray power, at room temperature. θ:2θ scans were performed between 10° and 80° with a step of 0.02°.

The oxide ζ-potential was measured using a Zetasizer Nano ZS (Malvern Instruments, Malvern, United Kingdom), operating at λ = 633 nm with a solid-state He-Ne laser at a scattering angle of 173°, using the dip-cell kit. Powders were suspended in a 0.01 M KNO_3_ aqueous solution. Each ζ-potential value was averaged from at least three measurements.

Thermogravimetric analyses (TGA) (Mettler Toledo, Milan, Italy) were performed on bare and modified HNT using a Mettler-Toledo TGA/DSC 3+ STAR System, between 30 and 900 °C with a rate of 3 °C/min under air flux.

Transmission electron microscopy (TEM) images were obtained with a Zeiss LEO 912ab Energy Filtering TEM (Zeiss, Oberkochen, Germany), equipped with a CCD-BM/1K system, operating at an acceleration voltage of 120 kV. Samples were suspended in 2-propanol or water (1 mg/mL) and deposited on Cu holey carbon grids (200 mesh).

Fourier transform infrared (FTIR) spectra were acquired using a Perkin Elmer Spectrum 100 spectrophotometer in Attenuated Total Reflectance (ATR) mode (Perkin Elmer, Milan, Italy), registering 12 scans between 4000 and 400 cm^−1^ with a resolution of 4.0 cm^−1^. A single-bounce diamond crystal was used with an incidence angle of 45°.

Diffuse reflectance spectra (DRS) were collected using a Shimadzu UV2600 spectrophotometer (Shimadzu, Milan, Italy).

### 2.4. PA6 and HNT Nanocomposites (PA6_HNT) Preparation and Characterization

#### 2.4.1. In Situ Polymerization

A 250 cm^3^ one-necked round bottom flask was loaded with ε-caprolactam (50.0 g), 6-aminocaproic acid (2.5 g) and the selected amount of HNT (either 1 or 4% w/w), and equipped with a polymerization candelabrum having nitrogen inlet and outlet adapters and an overhead mechanical stirrer. The flask was flushed with nitrogen, heated at 270 °C in a polymerization oven and then stirred (150–180 rpm) for 6 h. Afterwards, the reaction mixture was gradually brought back to room temperature. Flowing nitrogen atmosphere was maintained throughout the polymerization reaction. The resulting product was washed from unreacted ε-caprolactam with boiling water for 24 h, and dried in vacuum (about 4 mbar) at 100 °C for at least 12 h. For the sake comparison, neat PA6 was synthesized using the same conditions. 

#### 2.4.2. Melt Processing

Melt blended PA6-HNT nanocomposites (with filler content 1 or 4% w/w) were prepared using a ThermoScientific HAAKE Process 11 co-rotating twin screw extruder (Thermo Fisher Scientific, Rodano, Italy), with extrusion temperatures set at 235, 235, 235, 235, 235, 230, 217 and 210 °C from hopper to die respectively, a screw speed of 160 rpm, and a feed rate of 3 rpm. The extruded strands were quenched in water, air dried and then pelletized. Neat PA6, used as reference, was prepared by extruding PA6TECH under the same processing conditions. Prior to the use, nano-clays and PA6TECH pellets were dried under nitrogen atmosphere for 12 h at 60 °C in an oven to remove physisorbed water.

#### 2.4.3. PA6 Nanocomposites Characterization

The HNT actual content in PA6-nanocomposites was determined via dynamic TGA analyses performed from 30 to 900 °C at 10 °C/min, under nitrogen flow, using a Perkin Elmer TGA 4000 Instrument (Perkin Elmer, Milan, Italy). 

Scanning Electron Microscopy (SEM) studies of PA6 nanocomposites fracture surfaces were carried out on a Leica Electron Optics 435 VP microscope (Leica, Milan, Italy), working at an acceleration voltage of 15 kV and 50 pA of current probe and equipped with an energy dispersive X-ray (EDX) spectrometer (Philips, XL-30W/TMP), to investigate the HNT dispersion in PA6 polymer matrix. The samples were manually fractured after cooling in liquid nitrogen and sputter-coated with a 20 nm thick gold layer in rarefied argon, using an Emitech K550 Sputter Coater, with a current of 20 mA for 180 s.

The isothermal crystallization behavior of PA6 and its nanocomposites was determined using a Mettler Toledo DSC1 calorimeter (Mettler Toledo, Milan, Italy); analyses were conducted under nitrogen atmosphere, weighting 5–10 mg of each sample in a standard 40 µL aluminum crucible and using an empty 40 µL crucible as reference. 

In addition, non-isothermal behavior was studied using the following program: (i) heating at a rate of 10 °C/min from 25 to 250 °C; (ii) 5 min of isotherm at 250 °C; (iii) cooling from 250 to 25 °C at 10 °C/min (crystallization temperature, *T_c_* and crystallization enthalpy, Δ*H_c_*, were determined during cooling); (iv) 5 min of isotherm at 25 °C; and (v) heating from 25 to 250 °C at 10 °C/min (melting temperature, *T_m_*, and heat of fusion, Δ*H_f_*, were measured during heating). Δ*H_f_* was used to calculate the level of crystallinity, defined by the ratio of Δ*H_f_* to the heat of fusion of the purely crystalline forms of PA6 (Δ*H_f_*_°_). Since the Δ*H_f_*_°_ values of α and γ PA6 crystalline forms are nearly identical, the level of crystallinity was calculated using the average of the two, i.e., 240 J/g [35].

Rheological analyses, conducted using frequency sweep experiments, were performed with a Physica MCR 300 rotational rheometer with a parallel plate geometry (ϕ = 25 mm; 1 mm distance between plates) (Anton Paar, Rivoli, Italy). Neat PA6 and PA6-HNT nanocomposites were dried in a vacuum oven (around 4 mbar) at 100 °C for 12 h and linear viscoelastic regimes were studied; strain was set equal to 5% and curves of complex viscosity as function of frequency were recorded, taking 30 points ranging from 100 Hz to 0.1 Hz with a logarithmic progression, at 250 °C.

The molecular weight properties were evaluated using a Size Exclusion Chromatography (SEC) system having a Waters 1515 Isocratic HPLC pump, four Waters Styragel (103Å-104Å-105Å-500Å) columns and a UV detector (Waters 2487 Dual λ Absorbance Detector) at 244 nm, using a flow rate of 1 cm^3^·min^−1^ and 15 µL as injection volume (Waters, Sesto San Giovanni, Italy). Samples were prepared dissolving 10–15 mg of polymer in 1 cm^3^ of anhydrous DCM after N-trifluoroacetilation [43]; before the analysis, the solution was filtered with 0.45 µm filters. Molecular weight data were expressed in polystyrene (PS) equivalents. The calibration was built using monodispersed PS standards having the following nominal peak molecular weight (Mp) and molecular weight distribution (D): Mp = 1600 kDa (D ≤ 1.13), Mp = 1150 kDa (D ≤ 1.09), Mp = 900 kDa (D ≤ 1.06), Mp = 400 kDa (D ≤ 1.06), Mp = 200 kDa (D ≤ 1.05), Mp = 90 kDa (D ≤ 1.04), Mp = 50,400 Da (D = 1.03), Mp = 37,000 Da (D = 1.02), Mp = 17,800 Da (D = 1.03), Mp = 6520 Da (D = 1.03), Mp = 5460 Da (D = 1.03), Mp = 2950 Da (D = 1.06), Mp = 2032 Da (D = 1.06), Mp = 1241 Da (D= 1.07), Mp = 906 Da (D = 1.12); ethyl benzene (molecular weight = 106 g/mol). For all analyses, 1,2-dichlorobenzene was used as internal reference.

### 2.5. Hydrothermal Ageing Tests

Samples for the accelerated ageing test were prepared via injection molding with a BABYPLAST 610P instrument (BABYPLAST, Vicenza, Italy) and were immersed in distilled water at a set temperature of 90 °C for 70 days in glass jar containers. The main parameters of the injection molding are reported in Appendix A. The ageing temperature was chosen to accelerate the diffusion process as well as to accentuate the materials degradation mechanism [7,44]. Samples were fully immersed and periodically, i.e., after 25 (600 h), 50 (1200 h) and 70 days (1680 h), weighted using a digital balance of 0.0001 g accuracy. The moisture uptake (*M_t_*) was measured through a gravimetric method, as reported in Equation (1).
*M_t_* (%) = (*m_t_* − *m_o_*)/*m_o_* × 100(1)
where “*m_t_*” and “*m_o_*” are the wet and dry weights of the sample after any specific time *t*. Aged samples were analyzed via rheological, SEC and DSC analyses, according to the experimental procedures described in Section 2.4.3.

## 3. Results and Discussion

### 3.1. HNT Characterization and Functionalization

As previously stated, HNT_H_ and HNT_S_ fillers were acquired from different suppliers, and in order to investigate the differences between the two types of nanotubes, they were analyzed via SSA, XRD, TGA and TEM analyses. The two powders present different colors, HNT_S_ being white and HNT_H_ brownish (Appendix A). As reported in Table 1, HNT_S_ powder possesses a specific surface area of 53 m^2^/g, in good agreement with literature data [45,46]; on the other hand, HNT_H_ sample shows a lower surface area of 34 m^2^/g. According to XRD patterns reported in Figure 1A, HNT_S_ filler presents several impurity phases alongside halloysite, including kaolinite, quartz and gibbsite. These findings are in good agreement with previous literature reports on HNT samples from the same supplier [45,47]. Instead, HNT_H_ powder appears purer, as apparent from the absence or much lower intensity of impurity peaks, and more crystalline, as shown by the sharper halloysite (10Å) peaks. Our diffractogram is in good agreement with data from the supplier, showing only minor crystalline impurities, such as quartz (<1%) and kaolinite (<10%). 

TGA measurements are reported in Figure 1B. The initial mass loss, between 50 and 300 °C, can be associated with the removal of physisorbed and interlayer water, in this order. The most severe weight loss, at around 500 °C, can be attributed to the dehydroxylation of the inner surface [46]. The two sources of HNT show a slight difference in weight loss at lower temperatures. This can be related to the amount of physisorbed water on their surface: as the HNT_S_ sample has a higher surface area (Table 1), it also carries more water, resulting in a greater weight loss. Moreover, upon APTES addition a weight loss at around 250 °C can be appreciated, related to the degradation of the aminopropyl chain. Overall, in both cases loading is around 5% (Table 1), notwithstanding HNT_H_ lower surface area. This suggests that in this case functionalization was more efficient.

TEM images of untreated HNT were acquired to determine the difference in morphology between the two sources (Figure 1C). As can be seen in Table 1, the two families of HNT show greatly different aspect ratios: HNT_H_ appear much more elongated than HNT_S_, as their aspect ratio is three times larger, and also much less fragmented.

The FTIR spectra of HNT samples are reported in Figure 1D, in the 1800–1200 cm^−1^ range. As can be seen, HNT_S_ filler shows a more intense water bending peak at 1650 cm^−1^, which is in line with what reported by the TGA. After the functionalization with APTES, few signals between 1600 and 1300 cm^−1^ become appreciable: the peak at 1561 cm^−1^ can be attributed to –NH_2_ scissoring modes of APTES amino groups, while that at 1490 cm^−1^ can be related to the symmetric deformation of –NH_3_^+^ moieties. Finally, the band at 1335 cm^−1^ can be assigned to C–N stretching modes [48].

Interestingly, one of the most blatant differences between the two sources of HNT resides in their surface charge: as reported in Table 1, the recorded ς-potential at spontaneous pH for HNT_H_ is much more negative than that of HNT_S_, and this difference persists even after surface functionalization with APTES, which, due to its positively charged –NH_3_^+^ groups, raises the value of surface charge.

### 3.2. Synthesis of PA6 and PA6-HNT Nanocomposites via In Situ Polymerization and Melt Blending Processes

The dispersion of HNT_H_ and HNT_S_ inside PA6 polymer matrix was performed comparing two different incorporation routes, i.e., in situ polymerization and melt blending processes. The fillers were tested at 1 and 4% w/w, both unmodified and functionalized with APTES. Table 2 lists all of the prepared samples, along with the actual content of HNT inside the PA6 nanocomposites, as determined via TGA analyses. The latter are, in all cases, very close to the theoretical amount. 

It should be noted that at the highest filler content (4% w/w) HNT_H_ noticeably alters the original white color of neat PA6 (Appendix A), as also shown in DRS spectra (Appendix A).

To investigate the dispersion efficiency of HNT_H_ and HNT_S_ by the two preparation methods, the morphology of cryo-fractured PA6-HNT nanocomposite samples prepared was studied via SEM analyses; the presence of HNT-based aggregates was also investigated via EDX measurements. Representative images of 4% w/w composites are reported in Figure 2 and Figure 3.

In in situ polymerization samples, both unmodified and silane-modified HNT_S_ form micrometric aggregates (Figure 2C,G red circles), indicating that the filler has not been efficiently dispersed in the polymer matrix due to its poor interaction with PA6 [37]. On the other side, both pristine and functionalized HNT_H_ fillers are homogeneously dispersed in the polymer and no aggregates are present (Figure 2A,E). The reason behind this different behavior is probably the greater surface charge of HNT_H_, which more effectively separates nanotubes due to same-charge repulsing interactions. This effect, together with the higher aspect ratio leading to a more pronounced self-ordering, might favor a more homogeneous distribution of the filler in PA6_HNT_H_ composites, whereas HNT_S_ aggregates, possibly already present in the untreated powder, remain in the polymer matrix.

The dispersion of HNT_H_ further improves with APTES functionalization: APTES-modified HNT_H_ are enclosed and well-penetrated within the fractured PA6 surface. This observation can be explained considering that, while the surface charge remains high upon functionalization (Table 1), the amino functional groups of APTES promote the compatibility between the functionalized filler and PA6 matrix [49].

In the case of nanocomposites prepared via melt extrusion process (Figure 3), no significant difference related to the use of HNT_S_/HNT_H_ and APTES is detectable: in fact, the nanotubes distribution appears to be uniform across all the specimens and the good interaction between HNT and PA6 matrix seems to be independent of surface functionalization. However, it should be noted that in the case of melt blending, the HNT dispersion, although homogeneous, is essentially close to the surface, as also supported by the final performance (see Section 3.4 and Section 3.5). On the other hand, the in situ polymerization guarantees a bulk penetration of the filler. This difference may be explained by the fact that, as opposed to extrusion, during in situ preparation HNT fillers have more time to disperse in the medium, allowing the filler to spread evenly.

### 3.3. Isothermal Crystallization Behavior of PA6 Nanocomposites

The crystallization behavior of a crystalline polymer can be distinguished in two main processes, nucleation and crystal growth. The initial stage of crystallization, i.e., primary crystallization, can be analyzed by the Avrami kinetic equation (Equation (2)).
1 − *X*(*t*) = exp(−K·*t**^n^*)(2)
where *X*(*t*) is the fraction of crystallized polymer at time *t*, *n* is the parameter that details nucleation and growth processes, and K is the crystallization rate constant. Linear regression was adopted to obtain *n* and K from plots of the data in the form of Equation (3).
ln[1 − ln(1 − *X*(*t*))] = lnK + *n* ln*t*(3)

Based on the Avrami theory, *n* is related to the geometry of the crystal growth and on the type of nucleation. 

Here, isothermal crystallization behavior was studied by heating the sample from 25 to 250 °C at 10 °C/min, holding for 1 min to ensure melting, and then rapidly cooling to (i) 199 °C in the case of PA6_HNT_S_ nanocomposites prepared via in situ polymerization, (ii) 190 °C for PA6_HNT_H_ synthesized in situ and (iii) 200 °C for all samples prepared via melt processing. Neat PA6 was consequently analyzed at 190, 199 and 200 °C. The chosen crystallization temperatures (190/199/200 °C) were determined via a series of experiments conducted at various crystallization temperatures. Below the indicated onset crystallization temperature, the crystallization peak overlapped with the initial transient portion of heat flow curve. Setting the onset temperature as shown above deletes this problem, while it enables the completion of the crystallization process in a reasonable time. According to Fornes et al. [50], the different crystallization behavior is probably due to the different molecular weight of PA6 in the different composites. Moreover, in the case of PA6*_in situ_* nanocomposites, there can be an effect of the different interaction of the two used HNT with the polymer matrix, as previously described in SEM results. As reported in Table 2 (6th column), *n* values for pure PA6*_in situ_* and PA6_melt_ samples range from 1.6_T=190 °C_ to 1.8_T=199 and 200 °C_ and are consistent with values reported in the literature [51]. In PA6*_in situ_* nanocomposites, *n* shows a decrease with the addition of HNT_H_ filler and is not significantly affected by the presence of the functionalizing agent. According to Gurato et al. [52], this behavior suggests that small amounts of filler (lower than 5–10% w/w) can act as homogeneous nucleation sites followed by mono and bi-dimensional growth. On the other side, HNT_S_ samples overall show higher *n* values, possibly due to the not satisfactory HNT_S_ dispersion in PA6 matrix that enhances the formation of heterogeneous nucleation sites [52].

The *n* values for extruded materials are generally higher than the ones obtained by in situ polymerization. Also, in this case, functionalization with APTES does not significantly change the PA6 nanocomposites crystallization behavior. As shown by Khanna et al. [53], the increase in *n* can be related to memory effects associated with the processing of PA6. As a matter of fact, numerous studies have shown that the processing history of the polymer, e.g., melting, cooling, pelletizing and so on, are often not fully erased when the polymer is annealed at high melting temperatures [50,52,54]. In this way, memory effects associated with thermal and stress histories that remain present in the material after annealing in the melt may also lead to an increased formation of heterogeneous crystallization sites. 

The half-time of crystallization (*t*_1/2_) was calculated according to Equation (4):*t*_1/2_ = (ln(2/K))^1/*n*^(4)

The resulting values for all samples are reported in Table 2 (7th column). PA6*_in situ_* nanocomposites with bare and APTES-modified HNT_H_ fillers noticeably decrease their crystallization rate than neat PA6; this effect is due to the good adhesion between PA6 and the filler that impedes the motion of PA6 molecular chains, hampering the crystallization of PA6 matrix [13]. On the other hand, pristine and modified HNTs nanotubes lead to a less pronounced decrease of their crystallization rate with respect to neat PA6. Lastly, the crystallization rate of the samples obtained via melt extrusion process is not significantly influenced by the presence of bare and modified HNT. 

Overall, the results obtained in the case of samples prepared via melt blending are consistent with those reported in the literature [33,50]. On the basis of SEM results, the inefficient distribution of HNTs filler inside PA6 matrix by in situ polymerization results in a non-homogenous crystallization behavior of the corresponding composites. Moreover, the different dispersion of HNT_H_ during in situ polymerization and melt blending process is reflected in the crystallization behavior of the two types of composite: *n* and *t*_1/2_ are lower for the PA6*_in situ_* samples, consistently with the better filler incorporation within the bulk polymer matrix.

### 3.4. Non-Isothermal Crystallization Behavior of PA6 Nanocomposites

Non-isothermal crystallization studies were carried out to highlight the effect of pristine and modified fillers onto the crystallization behavior and crystal structure of the prepared PA6-HNT nanocomposites. Table 3 and Appendix A report the DSC data of PA6 nanocomposites in non-isothermal heating and cooling conditions.

By comparing the thermal behavior of each composite with respect to the relative neat PA6, a decrease in *T_c_* (Table 3, 2nd column) and Δ*H_c_* (Appendix A, 2nd column) values is observed upon addition of both pristine and APTES-modified nanotubes by in situ polymerization. In particular, *T_c_* values consistently show a decreasing trend at increasing HNT content. As described by Wurm et al. [55], the filler introduction in a polymer matrix hinders its macromolecular chains mobility, thus delaying crystal growth. Further evidence of slower kinetics for HNT_H_ and HNT_S_-based nanocomposites, can be seen in the χ_c_ values obtained upon subsequent heating (Table 3, 4th column). Δ*H_f_* values used for χ_c_ calculations are reported in the Appendix A (Appendix A, 3rd column). These effects are instead much less appreciable in the case of melt blending composites, in good agreement with isothermal crystallization studies (see Section 3.2) and with the literature [50,56].

Furthermore, it is interesting to note that in the case of the samples prepared via in situ and melt-blending processes, with 4% w/w of both pristine and modified HNT_H_/HNT_S_ the peak temperature related to the melting transition (*T_m_*) slightly decreases; the above observation may arise from several factors, such as the use of high filler loadings that could disturb the crystalline structures formation [35,57,58], but in particular, the formation of a small shoulder prior and close to the endothermic peak, associated with the PA6 γ crystalline form (Figure 4), may be a reason [59]. Several works [50,56,60,61] explain that the lower melting point associated with the presence of the γ form could be due to the lower crystalline density and increased melting entropy associated with the reduction in trans-conformation bonding as compared to the α counterpart. On the other hand, the presence of the γ crystalline form and the melting temperature decrease may simply reflect changes and imperfections in crystallites thickness and their distribution. In other words, the lower crystallinity could be due to the inability of polymer chains to be fully incorporated in to growing crystalline lamella; this leads to a smaller crystallite structures and more defects ridden crystalline lamella. Such imperfections in crystalline structure could contribute to the reduction of nanocomposites *T_m_* [62]. Furthermore, several works [33,35,56] report that HNT addition in polyamide-based composites favors polymer polymorphism, i.e., the formation of γ crystalline form. In particular, Guo et al. [56] suggested that the Si–O and –OH groups present along HNT surface are capable of forming hydrogen bonds with PA6 amide groups. The interfacial interaction could restrain the mobility of PA6 chains and weaken the hydrogen bonding interactions between the polymer chains, hindering the formation α-phase crystals.

### 3.5. Water Uptake and Rheological, SEC and DSC Characterization of Aged Nanocomposites

Figure 5 reports the absolute mass variation, calculated according to Equation (1), of PA6 and PA6-based nanocomposites conditioned in distilled water at 90 °C for 70 days. As widely reported in the literature [7,8,44,63], the PA6 hydrothermal stability can be studied at different ageing temperatures, e.g., at room temperature or close to water boiling point (~70–90 °C). It is generally accepted that temperature acts like an activator of eater diffusion within the polymer matrix [64]. As a result, a much shorter duration is needed to reach the equilibrium moisture content values at high temperature, such as 70–90 °C, than at room temperature [64]. Very recently, Sang et al. [8] compared the hydrothermal ageing at room temperature to that at temperatures higher than *T_g_*, showing similar trends in terms of water uptake at the different temperatures, although higher temperatures gave rise to a much faster initial water uptake and more appreciable surface and internal damage. Hence, in the present study, an ageing temperature of 90 °C was chosen to accelerate the diffusion processes that occur at lower ageing temperatures, as well as to accentuate degradation phenomena that otherwise would take weeks or months to happen [65,66].

The weight of aged samples after 25, 50 and 70 days of accelerated ageing test is reported in the Appendix A. The plotted data show the same trend involving two main steps, corresponding to the ageing periods of 25 and 50–70 days, respectively. In the first 25 days, an initial mass increase is observed as a consequence of a fast initial water uptake, associated with water penetration within PA6 amorphous regions; the mass increase continues until the equilibrium saturation level of the moisture uptake is reached [7]. Then, the equilibrium previously described disappears and a tendency of mass loss is noticed (25–70 days). The latter behavior can be ascribed to the polymer degradation consequent to hydrolysis phenomena [67]. 

The moisture uptake differs depending on the tested material. Overall, under the same ageing conditions, neat PA6 absorbs more water than modified PA6. Moreover, the decrease of moisture absorption gets higher with the increase of the modified-filler loading. This latter behavior is consistent with those reported in previous studies related to the combined use of functionalizing agents and polymers [68,69]. The improved decrease of water uptake in APTES-modified samples can be explained by the fact that amino functional groups present in the polymer matrix decrease PA6-HNT nanocomposites surface free energy [49]. Therefore, the use of modified-HNT fillers hinders and decelerates the mechanism of moisture sorption.

Furthermore, HNT_S_ seems to be more prone to water uptake than HNT_H_; as reported in Section 3.1, this behavior can be related to the HNT_S_ higher surface area that results in a greater capability of moisture sorption than HNT_H_ filler.

The effect of hydrothermal ageing on the rheological properties of PA6_HNT nanocomposites was studied. According to Appendix A, PA6*_in situ_* nanocomposites with the 4% w/w of modified filler have higher complex viscosity values than all other samples thanks to the combined effect of the higher HNT loading and APTES presence that favors the dispersion of the inorganic filler inside the polymer matrix [70]. In the case of PA6_melt_ nanocomposites (Appendix A), their complex viscosity is not significantly affected by the addition of bare and modified HNT.

After ageing, the complex viscosity decreases in all the samples prepared (Appendix A), with the exception of the PA6*_in situ_* samples prepared with the 4% of modified HNT (Figure 6). The complex viscosity decrease for PA6 and other polyamides after long ageing has been widely reported in the literature [71] and can be ascribed to the plasticization effect of water: during hydrothermal ageing, water molecules are mainly absorbed by the amorphous phase where they interfere with the hydrogen bonds and break the secondary bonds between the PA6 groups. This induces an increase in the chain mobility and decreases consequently the complex viscosity. However, Figure 6 clearly shows that the decrease of PA6*_in situ_* nanocomposites complex viscosities is only slightly present in the samples with modified HNT_H_/HNT_S_ at 4% w/w, in good agreement with water uptake results. The observed differenced in terms of complex viscosity between unaged and aged samples are not significant.

Furthermore, the effect of the hydrothermal ageing on PA6-based nanocomposites can be highlighted by the evaluation of the number average molecular weight ($\bar{Mn}$) and molecular weight distribution (D) determined by SEC analyses (Table 4). As described in Section 2.4.3, and in agreement with our previous works [9,18], samples were filtered with a 0.45 μm filter; therefore, molecular weight data refer only to the part of the polymer present in solution. In particular, no significant decrease in molecular weight can be observed in samples from melt blending pointing out to the absence of hydrolysis of the polymer chains due to the filler surface hydroxylation. The lack of any significant effect might be explained, on the one hand, with the drying step performed on the filler before incorporation, which removes physisorbed water that could lead to undesired hydrolysis reactions. On the other hand, the surface hydroxylation of HNT is mainly related to the inner lumen, exposing Al(OH)_x_ groups, while the outer surface of the nanotubes mainly exposes Si–O–Si groups and silanol groups are present only as defects [72]. The lack of accessible surface hydroxyl groups limits the occurrence of hydrolysis phenomena of the polymer matrix. However, in good agreement with water uptake behavior, a slight decrease in $\bar{Mn}$ data and broader D values are generally observed upon hydrothermal ageing as a result of degradation of PA6 chains [67]. This behavior is shown by both neat PA6 and HNT-PA6 composite, with the notable exception of the nanocomposites prepared with high loadings of modified fillers exhibiting almost unaltered parameters. For the latter samples, the sorption mechanism of water molecules is impeded by the presence of just 4% w/w of modified filler, which limits the diffusion of water molecules into the nanocomposites. As representative examples, Figure 7 compares the SEC curves obtained before and after the hydrothermal test of neat PA6 with those of PA6 nanocomposites prepared with the 4% w/w of modified filler.

Lastly, to evaluate the consequences of harsh ageing conditions on PA6 nanocomposites crystallization and melting temperatures, DSC measurements were obtained (Appendix A). Comparing the thermal data obtained before and after the ageing test, it is worth noting that only the following samples showed minor changes in crystallization kinetics:−PA6*_in situ_*_HNT_H__4_APTES: χ_c_unaged_: 14.0% vs. χ_c_aged_: 13.3%−PA6*_in situ_*_HNT_S__4_APTES: χ_c_unaged_: 14.9% vs. χ_c_aged_: 14.9%−PA6_melt__HNT_H__4_APTES: χ_c_unaged_: 21.7% vs. χ_c_aged_: 21.1%−PA6_melt__HNT_S__4_APTES: χ_c_unaged_: 19.3% vs. χ_c_aged_: 20.3%.

Together with the reported presence of PA6 γ crystalline even after the ageing test (Figure 8), this is further evidence of the high hydrothermal stability of the nanocomposites prepared with 4% w/w of modified HNT.

## 4. Conclusions

In the field of shelf-life improvement for polymer materials, the development of PA6 nanocomposites characterized by enhanced moisture absorption resistance is always in high demand due to the irreversible decay mechanisms subsequent to the accumulation of water molecules in the polymer matrix. In this context, the amino-silanization of HNT used as PA6 additives could be a way to create tailor-made materials with enhanced durability. 

The purpose of this study was to investigate the influence of very low amounts, i.e., 1% and 4% w/w, of HNT on the hydrothermal ageing resistance of PA6 nanocomposites. The use of very low filler loadings provides multiple advantages, including lower alteration of the polymer appearance and a less extended polymer/filler interface at which moisture accumulation may occur. 

Crystal growth and type of nucleation in PA6 nanocomposites were studied on the basis of Avrami theory via isothermal crystallization studies, and of dynamic DSC evaluations, showing that the use of low HNT loading does not influence the homogenous PA6 crystal growth and, at the same time, adding HNT promotes the PA6 polymorphism, i.e., the formation of γ counterpart. 

Here we employed a multifaceted approach, by studying the physicochemical properties of differently sourced HNT, the filler functionalization, and, last but not least, the dispersion technique, i.e., the in situ polymerization and melt extrusion processes. 

The HNT dispersion efficiency into the PA6 matrix was monitored by means of SEM/EDX analyses, showing clear differences between the two preparation methods: while HNT is homogeneously distributed via melt blending, in situ polymerization guarantees a better incorporation within the polymer matrix.

Both in situ and melt blending dispersion techniques showed that a lower surface area, a higher aspect ratio and a larger surface charge of HNT enhance filler incorporation and improve the final composite performances. It should be noted that the tested HNT samples also showed different structural properties in terms of crystallinity and phase impurity, which might affect the filler incorporation. These observations highlight the importance of a preliminary assessment of the physicochemical features of naturally sourced fillers, as relatively minor changes can notably affect the final composite properties. 

The filler surface modification with aminosilanes plays a major role in the durability of the PA6_HNT nanocomposites, as determined by very long hydrothermal ageing exposure (1680 h). Notably, the nanocomposites with 4% w/w of modified HNT_H_ and HNT_S_ appear to be the most durable ones in terms of water uptake and rheological, molecular weights, molecular weight distribution and thermal properties change before and after the ageing test. This might be related to the ability of amino groups present on HNT surface to avoid the diffusion of water molecules into the nanocomposites, enhancing the overall durability of the resulting nanocomposites.

The present research has led to a deep comprehension of the macromolecular, thermal and rheological properties of HNT-based nanocomposites that can be successfully adopted as composite materials with improved resistance to hydrothermal ageing. Future work will study the natural ageing of the most promising samples, i.e., PA6*_in situ_*_HNT_H__4_APTES and PA6*_in situ_*_HNT_S__4_APTES samples, in a real environment. 

## Figures and Tables

**Figure 1 polymers-12-00211-f001:**
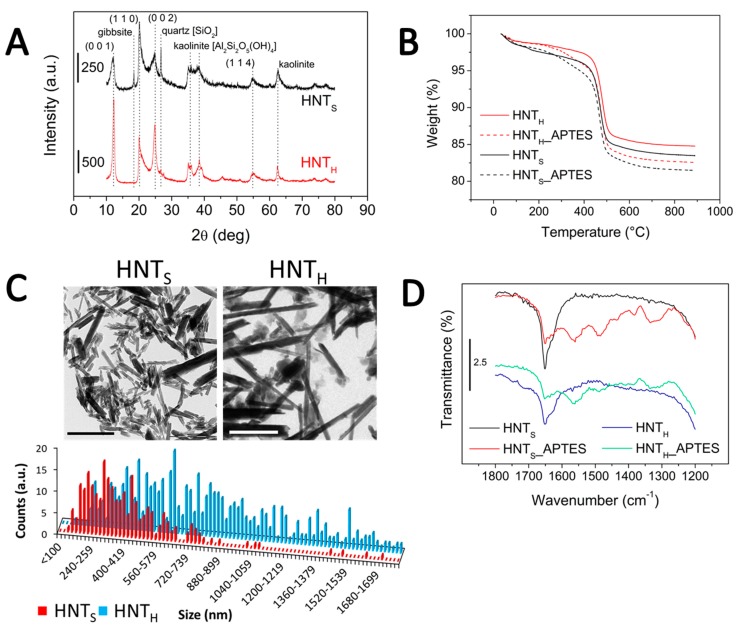
Characterization of adopted HNT powders: (**A**) XRD patterns with indexing of the halloysite peaks and main impurities; (**B**) TGA curves; (**C**) TEM images (scale bar: 1 μm); and (**D**) FTIR spectra of bare and APTES-modified fillers.

**Figure 2 polymers-12-00211-f002:**
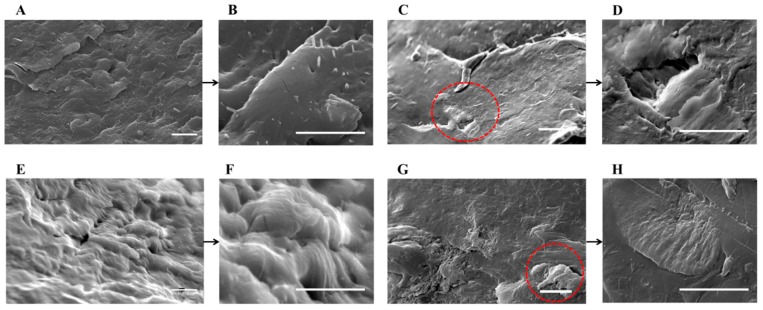
SEM images of (**A**,**B**) PA6*_in situ_*_HNT_H__4; (**C**,**D**) PA6*_in situ_*_HNT_S__4; (**E**,**F**) PA6*_in situ_*_HNT_H__4_APTES; and (**G**,**H**) PA6*_in situ_*_HNT_S__4_APTES (scale bar 2 µm).

**Figure 3 polymers-12-00211-f003:**
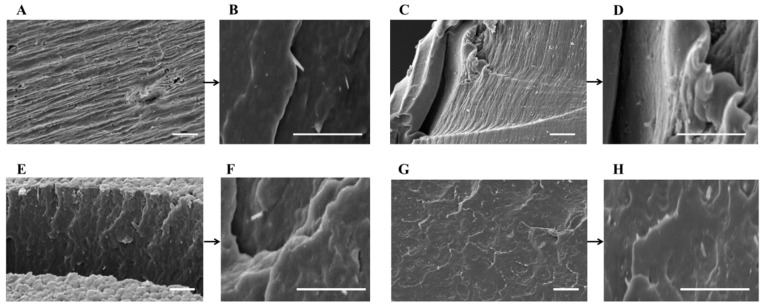
SEM images of (**A**,**B**) PA6_melt__HNT_H__4; (**C**,**D**) PA6_melt__HNT_S__4; (**E**,**F**) PA6_melt__HNT_H__4_APTES; and (**G**,**H**) PA6_melt__HNT_S__4_APTES (scale bar: 2 µm).

**Figure 4 polymers-12-00211-f004:**
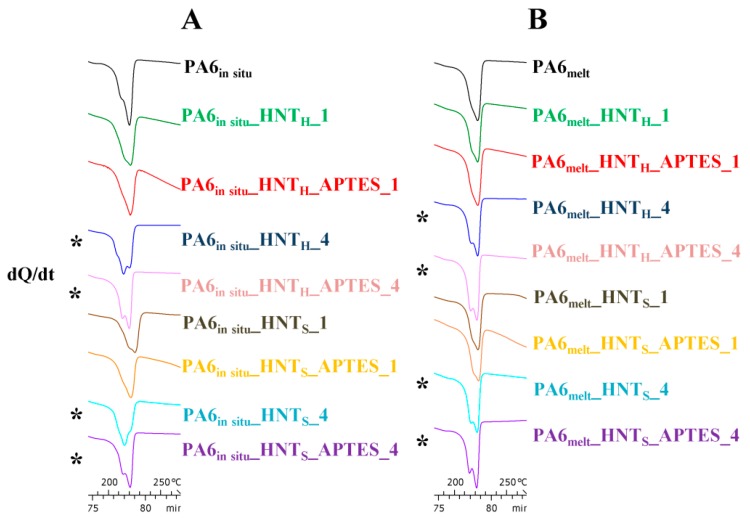
Heating scans of (**A**) PA6_HNT*_in situ_* and (**B**) PA6_HNT_melt_ nanocomposites. Samples characterized by the presence of γ form are highlighted with a star (*).

**Figure 5 polymers-12-00211-f005:**
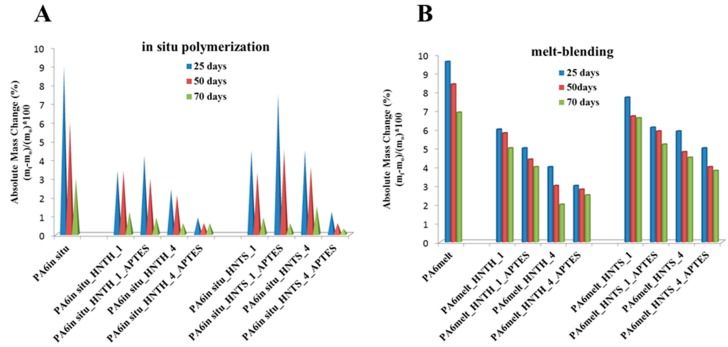
Water absorption in (**A**) PA6*_in situ_*_HNT and (**B**) PA6_melt__HNT nanocomposites.

**Figure 6 polymers-12-00211-f006:**
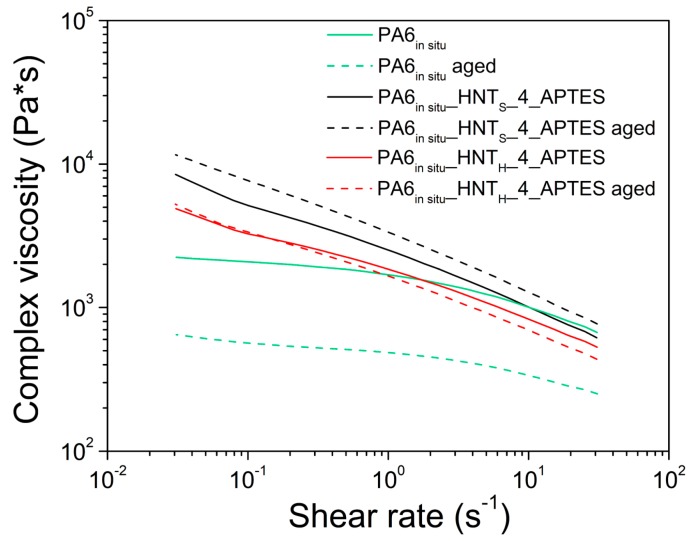
Rheological behavior of neat and APTES-functionalized HNT composites from in situ polymerization before and after hydrothermal ageing test.

**Figure 7 polymers-12-00211-f007:**
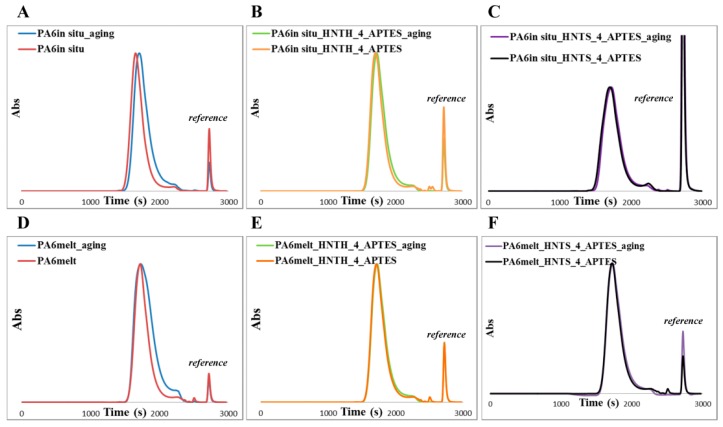
SEC curves before and after the ageing test of (**A**) PA6*_in situ_*, (**B**) PA6*_in situ_*_HNT_H__4_APTES, (**C**) PA6*_in situ_*_HNT_S__4_APTES, (**D**) PA6_melt_, (**E**) PA6_melt__HNT_H__4_APTES, and (**F**) PA6_melt__HNT_S__4_APTES nanocomposites.

**Figure 8 polymers-12-00211-f008:**
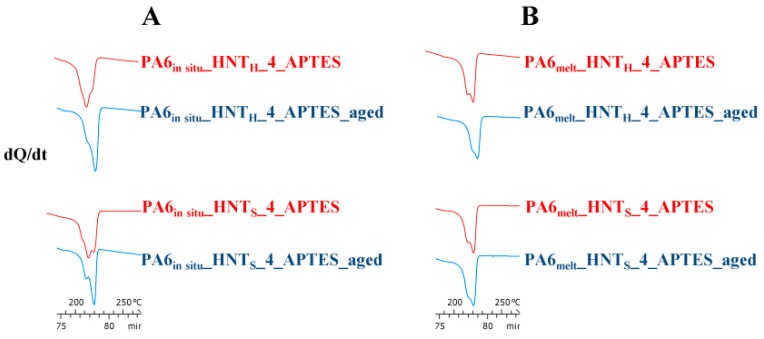
Heating scans of (**A**) PA6*_in situ_*_HNT, and (**B**) PA6*_melt_*_HNT nanocomposites prepared with the 4% w/w of modified HNT before and after the ageing test.

**Table 1 polymers-12-00211-t001:** Physicochemical properties of bare and functionalized HNT: specific surface area (SSA), average length and aspect ratio, ς-potential at spontaneous pH, and APTES loading from TGA.

Sample	SSA (m^2^/g)	Avg. Length (nm)	Avg. Aspect Ratio	ς-Potential (Bare HNT) (mV)	ς-Potential (HNT_APTES) (mV)	APTES Loading (%)
HNT_S_	53	400 ± 300	5 ± 3	−31 ± 2	+5.9 ± 0.6	4.6
HNT_H_	34	1100 ± 800	15 ± 9	−59 ± 3	−44 ± 2	5.1

**Table 2 polymers-12-00211-t002:** PA6 nanocomposites prepared via in situ polymerization and melt blending processes: theoretical and actual loading from TGA analyses, and kinetic analysis of isothermal crystallization data obtained via Avrami equation.

Preparation Route	Sample	HNT_theoretical_ (% w/w)	HNT_real_ (% w/w)	Avrami Data		
				onset *T_c_* (°C)	*n*	*t*_1/2_ (s)
*In situ* polymerization	PA6*_in situ_*	**-**	-	190	1.6	63
				199	1.8	408
	PA6*_in situ_*_HNT_H__1	1	0.8	190	1.0	135
	PA6*_in situ_*_HNT_S__1	1	0.9	199	2.3	408
	PA6*_in situ_*_HNT_H__4	4	3.3	190	1.1	163
	PA6*_in situ_*_HNT_S__4	4	4.0	199	1.6	408
	PA6*_in situ_*_HNT_H__1_APTES	1	0.7	190	1.0	137
	PA6*_in situ_*_HNT_S__1_APTES	1	1.0	199	2.4	420
	PA6*_in situ_*_HNT_H__4_APTES	4	4.0	190	1.2	174
	PA6*_in situ_*_HNT_S__4_APTES	4	3.9	199	1.6	408
Melt extrusion	PA6_melt_	-	-	200	1.8	135
	PA6_melt__HNT_H__1	1	1.0	200	2.3	135
	PA6_melt__HNT_S__1	1	0.6	200	2.6	135
	PA6_melt__HNT_H__4	4	4.0	200	2.3	135
	PA6_melt__HNT_S__4	4	4.0	200	2.1	135
	PA6_melt__HNT_H__1_APTES	1	0.8	200	2.3	126
	PA6_melt__HNT_S__1_APTES	1	0.7	200	2.1	126
	PA6_melt__HNT_H__4_APTES	4	3.8	200	2.0	135
	PA6_melt__HNT_S__4_APTES	4	3.6	200	2.0	135

**Table 3 polymers-12-00211-t003:** DSC data of PA6-HNT nanocomposites collected during cooling and second heating thermal steps.

Sample	Cooling	2nd Heating	
	*T_c_* (°C)	*T_m_* (°C)	χ_c_ (%)
PA6*_in situ_*	188.1	220.8	30.2
PA6*_in situ_*_HNT_H__1	187.2	220.8	23.8
PA6*_in situ_*_HNT_S__1	185.5	220.4	22.3
PA6*_in situ_*_HNT_H__4	182.1	219.0	23.5
PA6*_in situ_*_HNT_S__4	182.2	219.1	22.4
PA6*_in situ_*_HNT_H__1_APTES	187.2	220.3	21.5
PA6*_in situ_*_HNT_S__1_APTES	187.0	220.3	18.3
PA6*_in situ_*_HNT_H__4_APTES	184.6	214.6	14.0
PA6*_in situ_*_HNT_S__4_APTES	185.0	214.4	14.9
PA6_melt_	189.4	221.1	24.9
PA6_melt__HNT_H__1	188.8	221.2	20.1
PA6_melt__HNT_S__1	188.9	221.6	24.3
PA6_melt__HNT_H__4	186.2	220.1	17.1
PA6_melt__HNT_S__4	185.9	220.4	21.8
PA6_melt__HNT_H__1_APTES	188.6	221.4	24.4
PA6_melt__HNT_S__1_APTES	188.4	221.4	26.3
PA6_melt__HNT_H__4_APTES	187.1	219.8	21.7
PA6_melt__HNT_S__4_APTES	188.3	219.6	19.3

**Table 4 polymers-12-00211-t004:** Number average molecular weight ($\bar{Mn}$) and molecular weight distribution (D) data of PA6_HNT nanocomposites collected before and after the hydrothermal ageing test.

Sample	SEC_unaged_		SEC_aged_	
	$\bar{Mn}\;(\mathrm{Da})$	D	$\bar{Mn}\;(\mathrm{Da})$	D
PA6*_in situ_*	57,200	2.1	45,800	2.4
PA6*_in situ_*_HNT_H__1	48,600	2.1	32,400	2.5
PA6*_in situ_*_HNT_S__1	56,700	2.1	49,300	2.5
PA6*_in situ_*_HNT_H__4	60,400	2.0	42,000	2.2
PA6*_in situ_*_HNT_S__4	62,600	2.1	50,300	2.6
PA6*_in situ_*_HNT_H__1_APTES	55,600	2.1	45,400	2.4
PA6*_in situ_*_HNT_S__1_APTES	54,600	2.2	49,900	2.6
PA6*_in situ_*_HNT_H__4_APTES	47,800	2.3	41,800	2.3
PA6*_in situ_*_HNT_S__4_APTES	60,600	2.3	59,000	2.3
PA6_melt_	43,600	1.9	33,100	2.2
PA6_melt__HNT_H__1	40,300	2.0	19,300	2.5
PA6_melt__HNT_S__1	40,400	2.0	19,700	2.5
PA6_melt__HNT_H__4	42,400	2.0	21,000	2.3
PA6_melt__HNT_S__4	42,300	2.0	22,100	2.3
PA6_melt__HNT_H__1_APTES	40,400	2.0	33,100	2.2
PA6_melt__HNT_S__1_APTES	39,700	1.9	22,300	2.2
PA6_melt__HNT_H__4_APTES	42,500	2.0	42,000	2.0
PA6_melt__HNT_S__4_APTES	39,900	2.0	39,800	2.0

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
