# Peer review of "PA6 and Halloysite Nanotubes Composites with Improved Hydrothermal Ageing Resistance: Role of Filler Physicochemical Properties, Functionalization and Dispersion Technique"

_polymers, 2020, doi:10.3390/polym12010211_

Round 1

Reviewer 1 Report

The abstract section should be re-written to present the quantified data of the results. The tensile strength should be added to the results in order to compare in-situ polymerization and melt mixing method. In conclusion section please determine which type of nanocomposites has optimum performance In all cases of nanocomposites preparation it is obvious that the in-situ polymerization is better than melt mixing method. What is your objective to compare this two methods in the manuscript? Please more explain in last paragraph of introduction section.

Author Response

Response to Reviewer 1 Comments

We thank the Reviewer for his/her constructive comments that contributed to improve the overall quality of our manuscript. We have taken into consideration all the remarks and the manuscript was modified accordingly. A point by point answer to each Reviewer’s comment is reported in italic in the following text.

Reviewer 1 Comments:

Point 1: The abstract section should be re-written to present the quantified data of the results.

Response 1: In accordance with the Reviewer’s comment the Abstract (lines 19-35) was modified to include more quantitative details: “Polyamide 6 (PA6) suffers from fast degradation in humid conditions due to hydrolysis of amide bonds, which limits its durability. The addition of nanotubular fillers represents a viable strategy to overcome this issue, although the additive/polymer interface at high filler content can become privileged site for moisture accumulation. As a cost-effective and versatile material, halloysite nanotubes (HNT) were investigated to prepare PA6 nanocomposites with very low loadings (1-4% w/w). The roles of the physicochemical properties of two differently sourced HNT, of filler functionalization with (3-aminopropyl)triethoxysilane and of dispersion techniques (in situ polymerization vs. melt blending) were investigated. The aspect ratio (5 vs. 15) and surface charge (-31 vs. -59 mV) of the two HNT proved crucial in determining their distribution within the polymer matrix. In situ polymerization of functionalized HNT leads to enclosed and well-penetrated filler within the polymer matrix. PA6 nanocomposites crystal growth and nucleation type were studied according to Avrami theory, as well as the formation of different crystalline structures (α and γ). After 1680 hours of ageing, functionalized HNT reduced the diffusion of water into polymer, lowering water uptake after 600 h up to 90%, increasing the materials durability also regarding molecular weights and rheological behavior”.

Point 2: The tensile strength should be added to the results in order to compare in-situ polymerization and melt mixing method.

Response 2: We thank the Reviewer for the suggestion: we are currently investigating the mechanical properties of the materials obtained using DMTA. The complete mechanical characterization will be the subject of a following paper, since the present one is focused on the dispersion efficiency of HNT and on the resulting composite hydrothermal stability. This made the present paper quite lengthy and complex, including a broad range of characterization techniques and preparation procedures. Hence, we think that this further characterization will deserve a full paper.

Point 3: In conclusion section please determine which type of nanocomposites has optimum performance.

Response 3: In agreement with the Reviewer’s remark, we have better highlighted in the Conclusions section that PA6in situ nanocomposites with 4% w/w of APTES-modified HNTH/S are the more promising samples (lines 523-525 and 531-532): “Notably, the nanocomposites with 4% w/w of modified HNTH and HNTS appear to be the most durable ones in terms of water uptake and rheological, molecular weights, molecular weight distribution and thermal properties change before and after the ageing test.”;“Future work will study the natural ageing of the most promising samples, i.e. PA6in situ_HNTH_4_APTES and PA6in situ_HNTS_4_APTES samples, in a real environment.”

Point 4: In all cases of nanocomposites preparation it is obvious that the in-situ polymerization is better than melt mixing method. What is your objective to compare this two methods in the manuscript? Please more explain in last paragraph of introduction section.

Response 4: In the preparation of our PA6 nanocomposites, we have observed that the two adopted preparation procedures differ in the role of HNT surface charge with respect to the filler dispersion efficiency. In particular, during in situ polymerization, a homogeneous dispersion inside the polymer matrix of unmodified HNTs was quite difficult to attain due to its surface charge, resulting in the formation of aggregates (Figure 2); this phenomenon influences the final nanocomposites properties, in particular the PA6 crystallization behavior and polymorphism. In agreement with this observation, we have revised the last paragraph of the Introduction (lines 106-108): “In particular, the comparison between in situ and melt blending methods has highlighted the role of HNT surface charge with respect to the filler dispersion efficiency.”.

Reviewer 2 Report

The papers deals with 2 kinds of halloysites, but not enough information is given about these materials. in fact, these two materials are very different since the HNTs is a carefully selected halloysite from Dragon mine (Utah, USA) which contains few impurities whereas these of iMinerals (also from USA but from Idaho, it seems) contains 30% kaolinite from the website of the company:

https://www.imineralsinc.com/our-products/halloysite

Hence, a lot of differences about the reactivity could be due to the nature of minerals and not only due to the different values of specific surface area. The authors should indicate the exact composition of minerals used, with the use of X-Fluorescence or the supply of accurate information.

P2  melt intercalation is mentioned. Nevertheless, for halloysite, due to its very low interlamellar space, this expression is not relevant.

About the SEM pictures, please avoid distortion and use a square format. In some case, aggregates are observed, but some bundles or HNT could remain after grinding of the material from the deposit.

About the study of the crytsllization process, the selection of the temperatures of 190, 199 and 200°C should be justified.

For some compositions Mn increases for some composites in comparison with this of pristine in situ PA6. This effect should be explained. Moreover it seems surprising not to observe a significant decrease of Mn for PA6melt with the incorporation of HNT, due to hydrolysis caused by the hydroxyl groups present at the surface of the mineral.

An increase of complex viscosity is noticed for PA6 in situ HNTs 4 APTES aged. How this phenomenon can be explained? It can be noticed that in this cas the Mn has decreased.

The choice of the temperature of 90°C is completely arbitrary, not justified in the article and it appears difficult to draw information about the behaviour at other temperatures;  The interest of this procedure of accelerated test is not enough interesting if natural ageing is not also performed.

The weight of aged samples is not indicated. It is really surprising that the relative uptake of water is lower for 70 days in comparison with 50 or 25 days.

Author Response

Response to Reviewer 2 Comments

We wish to thank the Reviewer for his/her fruitful suggestions. We believe that, thanks to his/her contribution, the overall quality of our manuscript has now been improved. All of the raised points have been carefully considered and modifications have been introduced in the manuscript accordingly. A point-by-point answer is reported in italic in the following.

Reviewer 2 Comments:

Point 1: The papers deals with 2 kinds of halloysites, but not enough information is given about these materials. In fact, these two materials are very different since the HNTs is a carefully selected halloysite from dragon mine (Utah, USA) which contains few impurities whereas these of iMinerals (also from USA but from Idaho, it seems) contains 30% kaolinite from the website of the company:

https://www.imineralsinc.com/our-product/halloysite

Hence, a lot of differences about the reactivity could be due to the nature of minerals and not only due to the different values of specific surface area. The authors should indicate the exact composition of minerals used, with the use of X-Fluorescence or the supply of accurate information.

Response 1: As the Reviewer correctly pointed out, halloysite nanotubes suffer from a large variability in several physicochemical properties depending on the mine of origin. We are aware of this issue and that is why we decided to compare different commercial halloysite materials. However, according to our X-ray diffraction characterization, as well as to the technical sheet from the suppliers, HNTH has a higher purity in terms of phase composition with respect to HNTs. In particular, a kaolinite content of 9.5% and a quartz amount <1% is observed in HNTH, which is known with the commercial name Ultra HalloPure (which was incorrectly identified as HalloPure in the previous version of the manuscript, line 121) and has a higher purity that other products from the same mine. This aspect was better clarified in the text (lines 229-236: “According to XRD patterns reported in Figure 1A, HNTS filler presents several impurity phases alongside halloysite, including kaolinite, quartz and gibbsite. These findings are in good agreement with previous literature reports on HNT samples from the same supplier [45,47]. Instead, HNTH powder appears purer, as apparent from the absence or much lower intensity of impurity peaks, and more crystalline, as shown by the sharper halloysite-(10Å) peaks. Our diffractogram is in good agreement with data from the supplier, showing only minor crystalline impurities, such as quartz (< 1%) and kaolinite (<10%).” Notwithstanding the importance of the structural variability between the two HNT samples, we believe that the morphological and surface charge differences play a major role on the incorporation within the polymer matrix and on the final composite performance. In partial acceptance of the Reviewer remark, we added a note of caution in the Conclusion section (lines 516-520): “Both in situ and melt intercalation blending dispersion techniques showed that a lower surface area, a higher aspect ratio and a larger surface charge of HNT enhance filler incorporation and improve the final composite performances. It should be noted that the tested HNT samples showed as well different structural properties in terms of crystallinity and phase impurity, which might affect the filler incorporation. These observations highlight the importance of a preliminary assessment of the physicochemical features of naturally sourced fillers, as relatively minor changes can notably affect the final composite properties.”

Point 2: P2 melt intercalation is mentioned. Nevertheless, for halloysite, due to its very low interlamellar space, this expression is not relevant.

Response 2: In agreement with the Reviewer’s remark, we have changed the expression “melt intercalation” with “melt blending/melt extrusion”. Please, see the revised manuscript: lines 27, 87, 89, 379, 509 and 514.

Point 3: About the SEM pictures, please avoid distortion and use a square format. In some case, aggregates are observed, but some bundles or HNT could remain after grinding of the material from the deposit.

Response 3: We have revised the presentation of SEM figures according to the Reviewer’s request. Please, see the revised Figures 2 and 3. We agree with the Reviewer that some degree of aggregation could be present in the untreated HNT material. However, the preparation procedure seems to play a more important role on the aggregation behavior of HNT in composite samples observed in SEM images; as a matter of fact, the same filler (e.g. HNTs) is present either in a well dispersed form or as aggregated bundles depending on the adopted preparation procedure. In accordance with the Reviewer’s suggestion, the following paragraph was modified in Section 3.2 (lines 288-291): “This effect, together with the higher aspect ratio leading to a more pronounced self-ordering, might favor a more homogeneous distribution of the filler in PA6_HNTH composites, whereas HNTS aggregates, possibly already present in the untreated powder, remain in the polymer matrix.”

Point 4: About the study of the crytsllization process, the selection of the temperatures of 190, 199 and 200°C should be justified.

Response 4: In order to determine the isothermal crystallization behavior of PA6 nanocomposites, samples were heated from 25 to 250°C at 10°C/min, held for 1 min to ensure melting, then rapidly cooled to 190°C in the case of PA6in situ_HNTH samples, 199°C for PA6in situ_HNTS nanocomposites and 200°C for PA6melt samples, where they were allowed to crystallize. The chosen crystallization temperature (190/199/200°C) was determined via a series of experiments conducted at various temperatures. Below the selected temperature, the onset of crystallization overlapped with the initial transient portion of the heat flow curve. Setting the crystallization temperature as indicated eliminated this problem, enabling a complete detection of the crystallization phenomenon without interference from the beginning of the isothermal DSC analysis. According to Fornes et al. (reference 50, doi:10.1016/S0032-3861(03)00344-6), the different obtained crystallization temperatures are probably due to the different molecular weights of the polymer matrix. Moreover, in the case of PA6in situ nanocomposites, this effect can be also related to the different dispersion efficiency of HNTS/HNTH with ensuing different interaction between the filler and the polymer matrix. To better clarify this point, the corresponding paragraph was modified as follows (lines 325-326 and 329-333): “The chosen crystallization temperatures (190/ 199/ 200 °C) were determined via a series of experiments conducted at various crystallization temperatures.” “According to Fornes et al. [50], the different crystallization behavior is probably due to the different molecular weight of PA6 in the different composites. Moreover, in the case of PA6in situ nanocomposites, there can be an effect of the different interaction of the two used HNT with the polymer matrix, as previously described in SEM results.”

Point 5: For some compositions Mn increases for some composites in comparison with this of pristine in situ PA6. This effect should be explained. Moreover it seems surprising not to observe a significant decrease of Mn for PA6melt with the incorporation of HNT, due to hydrolysis caused by the hydroxyl groups present at the surface of the mineral.

Response 5: As reported in Table 4, Mn values of PA6in situ_HNTH and PA6in situ_HNTS are very close and present the very similar distribution, hence we do not consider the slight increase observed in a few cases to be significant. Moreover, the authors are grateful to the Reviewer for the insightful comment about the possible effect of surface hydroxylation of HNT on PAmelt samples. First of all, it is important to underline that SEC data refer only to the polymer that is not bonded to HNT, which is retained by the filter during sample preparation for the analysis. Furthermore, the lack of any significant effect might be explained, on the one hand, with the drying step performed on the filler before incorporation, which removes physisorbed water that could lead to hydrolysis reactions. On the other hand, the surface hydroxylation of HNT is mainly related to the inner lumen, exposing Al(OH)x groups, while the outer surface of the nanotubes mainly exposes Si-O-Si groups and silanol groups are present only as defects (see e.g. reference 68 DOI:10.1016/j.clay.2015.05.001). The lack of accessible surface hydroxyl groups limits the occurrence of hydrolysis phenomena of the polymer matrix. A paragraph was added in Section 3.5 to comment on this point (lines 453-464): “As described in Section 2.4.3 and in agreement with our previous works (9, 18), samples were filtered with a 0.45 μm filter, therefore molecular weight data refer only to the part of the polymer present in solution. In particular, no significant decrease in molecular weight can be observed in samples from melt blending pointing out to the absence of hydrolysis of the polymer chains due to the filler surface hydroxylation. The lack of any significant effect might be explained, on the one hand, with the drying step performed on the filler before incorporation, which removes physisorbed water that could lead to undesired hydrolysis reactions. On the other hand, the surface hydroxylation of HNT is mainly related to the inner lumen, exposing Al(OH)x groups, while the outer surface of the nanotubes mainly exposes Si-O-Si groups and silanol groups are present only as defects [68]. The lack of accessible surface hydroxyl groups limits the occurrence of hydrolysis phenomena of the polymer matrix.”

Point 6: An increase of complex viscosity is noticed for PA6 in situ HNTs_4_APTES aged. How this phenomenon can be explained? It can be noticed that in this case the Mn has decreased.

Response 6: Out of clarity, in the following table we report the complex viscosity data of the unaged and aged PA6in situ_HNTS_4_APTES samples shown in Figure 6. The rheological behavior of unaged and aged samples is very similar, and we do not think the small complex viscosity increase observed in the case of the aged sample is relevant. Out of clarity, we added the following comment in the text (lines 445-446): “The observed differences in terms of complex viscosity between unaged and aged samples are not significant.”

Furthermore, as previously said in Point 5 and in according with our previous works (references 9 and 18), SEC data are referred only to the part of the polymer present in solution, so a direct correlation between rheological and molecular weight behavior cannot be performed.

PA6_HNTS_4_APTES

Unaged

PA6_HNTS_4_APTES

Aged

Shear Rate

Complex Viscosity

Shear Rate

Complex Viscosity

[1/s]

[Pa·s]

[1/s]

[Pa·s]

3.06E+01

6.16E+02

3.08E+01

7.69E+02

2.51E+01

6.82E+02

2.49E+01

8.53E+02

1.92E+01

7.56E+02

1.92E+01

9.48E+02

1.51E+01

8.40E+02

1.52E+01

1.05E+03

1.20E+01

9.32E+02

1.23E+01

1.17E+03

9.51E+00

1.03E+03

9.61E+00

1.30E+03

7.58E+00

1.14E+03

7.61E+00

1.44E+03

5.99E+00

1.26E+03

6.09E+00

1.60E+03

4.73E+00

1.39E+03

4.79E+00

1.77E+03

3.72E+00

1.53E+03

3.76E+00

1.96E+03

2.93E+00

1.68E+03

2.96E+00

2.17E+03

2.30E+00

1.85E+03

2.32E+00

2.39E+03

1.81E+00

2.02E+03

1.83E+00

2.64E+03

1.42E+00

2.22E+03

1.44E+00

2.90E+03

1.12E+00

2.42E+03

1.13E+00

3.20E+03

8.79E-01

2.63E+03

8.87E-01

3.51E+03

6.91E-01

2.86E+03

6.98E-01

3.85E+03

5.43E-01

3.11E+03

5.49E-01

4.22E+03

4.27E-01

3.36E+03

4.31E-01

4.61E+03

3.36E-01

3.62E+03

3.39E-01

5.03E+03

2.64E-01

3.90E+03

2.67E-01

5.48E+03

2.08E-01

4.18E+03

2.10E-01

5.96E+03

1.63E-01

4.48E+03

1.65E-01

6.47E+03

1.28E-01

4.80E+03

1.30E-01

7.03E+03

1.01E-01

5.13E+03

1.02E-01

7.62E+03

7.89E-02

5.58E+03

8.01E-02

8.28E+03

6.22E-02

6.16E+03

6.29E-02

8.99E+03

4.95E-02

6.83E+03

4.92E-02

9.81E+03

3.91E-02

7.57E+03

3.88E-02

1.07E+04

3.05E-02

8.46E+03

3.05E-02

1.16E+04

Point 7: The choice of the temperature of 90°C is completely arbitrary, not justified in the article and it appears difficult to draw information about the behavior at other temperatures. The interest of this procedure of accelerated test is not enough interesting if natural ageing is not also performed.

Response 7: The ageing temperature of 90°C was chosen to accelerate the diffusion process as well as to accentuate the materials degradation mechanism, as also widely reported in the literature (see e.g. reference 7, and new reference 44). We chose 90°C also because accelerated ageing for a series of “industrial” tests on polyamide is usually set at a minimum of 80°C: the main reason underlying the choice of 80°C is that this temperature is over Tg of the material and therefore allows to accelerate phenomena that otherwise would take weeks or months to occur. In the Materials & Methods section, we have added a further relevant reference to corroborate the adopted procedure (line 215).

Some works, such as:

i) Lin Sang, Chuo Wang, Yingying Wang, Wenbin Hou, Effects of hydrothermal aging on moisture absorption and property prediction of short carbon fiber reinforced polyamide 6 composites, Composites Part B: Engineering, Volume 153, 2018, Pages 306-314, ISSN 1359-8368, https://doi.org/10.1016/j.compositesb.2018.08.138;

ii) Lei, Y., Zhang, J., Zhang, T., & Li, H. (2019). Water diffusion in carbon fiber reinforced polyamide 6 composites: Experimental, theoretical, and numerical approaches. Journal of Reinforced Plastics and Composites, 38(12), 578-587. https://doi.org/10.1177/0731684419835034;

studied the PA6 hydrothermal stability at different temperatures, and from the comparison of the data obtained it is clear that, as the ageing temperature decreases, the diffusion process gets slower; the ageing temperature used in our work enables to gather data about the hydrothermal stability in a reasonable time. We agree with the Reviewer that a comparison between accelerated and natural ageing could provide useful information. However, our approach, also in other research areas (see e.g. Sabatini et al. Polymers, (2019), DOI: 10.3390/polym11071190), is based on screening the physicochemical stability of the new materials prepared via accelerated ageing test, and only the best performing ones are then studied in a real environment, which requires much longer time. In subsequent works, we will report the natural hydrothermal ageing of PA6in situ_HNTH/S_4_APTES samples, which is currently under way. We have added a short sentence about prospective works in the Conclusions section (lines 531-532): “Future work will study the natural ageing of the most promising samples, i.e. PA6in situ_HNTH_4_APTES and PA6in situ_HNTS_4_APTES samples, in a real environment”.

Point 8: The weight of aged samples is not indicated. It is really surprising that the relative uptake of water is lower for 70 days in comparison with 50 or 25 days.

Response 8: The authors wish to thank the Reviewer for the useful remark. Data plotted in Figure 5 are relative to the absolute mass change calculated via Equation 1. In accordance with the cited literature (references 7 and 63), mass change data show a fast initial water uptake, associated with the water penetration within PA6 amorphous regions, which continues until the equilibrium saturation level of the moisture uptake is reached (ageing period: 0-25 days). Then, in the ageing time corresponding to 25-70 days, the equilibrium previously described disappears and a tendency of mass loss is observed. As reported by Chaupart et al. (reference 63, doi:10.1016/S0032-3861(97)00414-X), this latter behavior is due to the polymer degradation due to hydrolysis phenomena. Out of clarity, we have better clarified the commenting text in Section 3.5 (lines 407-416) and we have added a table in Supporting Information (new Table S3) reporting the weight of aged samples: “Figure 5 reports the absolute mass variation, calculated in according with Equation 1, of PA6 and PA6-based nanocomposites conditioned in distilled water at 90°C for 70 days. The weight of aged samples after 25, 50 and 70 days of accelerated ageing test is reported in Supporting Information, Table S3. The plotted data show the same trend involving two main steps, corresponding to the ageing periods of 25 and 50-70 days, respectively. In the first 25 days, an initial mass increase is observed as a consequence of a fast initial water uptake, associated with water penetration within PA6 amorphous regions; the mass increase continues until the equilibrium saturation level of the moisture uptake is reached [7].Then, the equilibrium previously described disappears and a tendency of mass loss is noticed (25-70 days). The latter behavior can be ascribed to the polymer degradation consequent to hydrolysis phenomena [63].”.

Round 2

Reviewer 1 Report

Thank you for the corrections in your manuscript bases on my comments

Author Response

Response to Reviewer 1 Comments

Dear reviewer 1, We have been honored to read your opinion about our research. Thank you for the time and attention given to our work.

Reviewer 1 Comments:

Thank you for the corrections in your manuscript bases on my comments.

Reviewer 2 Report

I consider that the most part of remarks were taken into account. I appreciated the answers and comments following my questions and remarks as well as the most part of the modifications carried out.

However, further  revision is needed for the present version despite the asked modifications.

The main point is always about the selection of the temperature of 90°C for the accelerated ageing tests. This choice should be fully justified in the text and not only by just giving 2 aditional references. It is important to mention and assess if the value of 90°C corresponds really to an acceleration of the same phenomena or corresponds to different ageing phenomena than these observed at lower temperatures. The chemical processes as well as the Tg should be considered. These elements should be developed in the manuscript by using the available literature.

The second point corresponds to the explanations given about the decrease of moculecular weight. There is a problem of understanding for the incomplete sentence L. 494-495.

Author Response

Response to Reviewer 2 Comments

Reviewer 2 Comments:

I consider that the most part of remarks were taken into account. I appreciated the answers and comments following my questions and remarks as well as the most part of the modifications carried out. However, further revision is needed for the present version despite the asked modifications.

Response: We wish to thank the Reviewer’s for his/her constructive comments that contributed to improve the overall quality of our manuscript. We have taken into consideration all the remarks and the manuscript was modified accordingly. The answers to Reviewer’s comments will appear in red in the following text.

Point 1: The main point is always about the selection of the temperature of 90°C for the accelerated ageing tests. This choice should be fully justified in the text and not only by just giving 2 aditional references. It is important to mention and assess if the value of 90°C corresponds really to an acceleration of the same phenomena or corresponds to different ageing phenomena than these observed at lower temperatures. The chemical processes as well as the Tg should be considered. These elements should be developed in the manuscript by using the available literature.

Response 1: According to the Reviewer’s remark, we have better motivated the choice of an aging temperature of 90°C for the PA6-HNT nanocomposites hydrothermal aging, also on the grounds of the literature (refs. 7, 8 and 44, already cited in the previous version of the manuscript, and new refs. 63-64-65-66-67):

-reference 7: Ksouri, I.; De Almeida, O.; Haddar, N. Long term ageing of polyamide 6 and polyamide 6 reinforced with 30% of glass fibers: physicochemical, mechanical and morphological characterization. J. Polym. Res. 2017, 24.

-reference 8: Sang, L.; Wang, C.; Wang, Y.; Hou, W. Effects of hydrothermal aging on moisture absorption and property prediction of short carbon fiber reinforced polyamide 6 composites. Compos. Part B Eng. 2018, 153, 306–314.

-reference 44: Piao, H. Influence of water absorption on the mechanical properties of discontinuous carbon fiber reinforced polyamide 6. J. Polym. Res. 2019.

-reference 63: Lin Sang, Chuo Wang, Yingying Wang, Wenbin Hou. Effects of hydrothermal aging on moisture absorption and property prediction of short carbon fiber reinforced polyamide 6 composites. Compos. Part B Eng. 2018, 153, 306–314.

-reference 64: Lei, Y.; Zhang, J.; Zhang, T.; Li, H. Water diffusion in carbon fiber reinforced polyamide 6 composites: Experimental, theoretical, and numerical approaches. J. Reinf. Plast. Compos. 2019, 38, 578–587.

-reference 65: Taktak, R.; Guermazi, N.; Derbeli, J.; Haddar, N. Effect of hygrothermal aging on the mechanical properties and ductile fracture of polyamide 6: Experimental and numerical approaches. Eng. Fract. Mech. 2015, 148, 122–133.

-reference 66: Lihua Gao, Lin Ye, G.L. Long-Term Hydrothermal Aging Behavior and Life-Time Prediction of Polyamide 6. J. Macromol. Sci. Part B 2015, 54, 239–252.

-reference 67: Ruiguang Li, Lin Ye, G.L. Long-Term Hydrothermal Aging Behavior and Aging Mechanism of Glass Fibre Reinforced Polyamide 6 Composites. J. Macromol. Sci. Part B 2017, 57, 67–82.

As widely reported in the literature, the used aging temperature, over to the PA6 Tg, allows to accentuate and therefore accelerate the degradation processes that happen at lower ageing temperatures. A commenting paragraph was added to Section 3.5 (lines 404-415): “As widely reported in the literature [7,44,63,64], the PA6 hydrothermal stability can be studied at different ageing temperatures, e.g. at room temperature or close to water boiling point (~70-90°C). It is generally accepted that temperature acts like an activator of eater diffusion within the polymer matrix [65]. As a result, a much shorter duration is needed to reach the equilibrium moisture content values at high temperature, like 70-90°C, than at room temperature [65]. Very recently, Sang et al. [8] compared the hydrothermal aging at room temperature to that at temperatures higher than Tg, showing similar trends in terms of water uptake at the different temperatures, although higher temperatures gave rise to a much faster initial water uptake and more appreciable surface and internal damage. Hence, in the present study, an ageing temperature of 90°C was chosen to accelerate the diffusion processes that occur at lower ageing temperatures as well as to accentuate degradation phenomena that otherwise would take weeks or months to happen [66,67].”.

Point 2: The second point corresponds to the explanations given about the decrease of moculecular weight. There is a problem of understanding for the incomplete sentence L. 454-455.

Response 2: We apologize for the mistake made while reviewing the previous version of our manuscript. We have corrected the corresponding text. Please, see in the main text lines 459-470: “As described in Section 2.4.3 and in agreement with our previous works [9,18], samples were filtered with a 0.45 μm filter, therefore molecular weight data refer only to the polymer present in solution. In particular, no significant decrease in molecular weight can be observed in samples from melt blending pointing out the absence of hydrolysis of the polymer chains due to the filler surface hydroxylation. The lack of any significant effect might be explained, on the hand, with the drying step performed on the filler before incorporation, which removes physisorbed water that could lead to undesired hydrolysis reactions. On the other hand, the surface hydroxylation of HNT is mainly related to the inner lumen, exposing Si-O-Si groups and silanol groups are present only as defects [68]. The lack of accessible surface hydroxyl groups limits the occurrence of hydrolysis phenomena of the polymer matrix.

Round 3

Reviewer 2 Report

Thnak you for your efforts about this revision